# Probabilistic Autoencoder

**Vanessa Böhm**                                                              *vboehm@berkeley.edu*
*Berkeley Center for Cosmological Physics*
*Department of Physics*
*University of California*
*Berkeley, CA, USA*
*Lawrence Berkeley National Laboratory*

**Uroš Seljak**                                                              *useljak@berkley.edu*
*Berkeley Center for Cosmological Physics*
*Department of Physics*
*University of California*
*Berkeley, California, USA*
*Lawrence Berkeley National Laboratory*

**Reviewed on OpenReview:** *https://openreview.net/forum?id=AEoYjvjKVA*

## Abstract

Principal Component Analysis (PCA) minimizes the reconstruction error given a class of linear models of fixed component dimensionality. Probabilistic PCA adds a probabilistic structure by learning the probability distribution of the PCA latent space weights, thus creating a generative model. Autoencoders (AE) minimize the reconstruction error in a class of nonlinear models of fixed latent space dimensionality and outperform PCA at fixed dimensionality. Here, we introduce the Probabilistic Autoencoder (PAE) that learns the probability distribution of the AE latent space weights using a normalizing flow (NF). The PAE is fast and easy to train and achieves small reconstruction errors, high sample quality, and good performance in downstream tasks. We compare the PAE to Variational AE (VAE), showing that the PAE trains faster, reaches a lower reconstruction error, and produces good sample quality without requiring special tuning parameters or training procedures. We further demonstrate that the PAE is a powerful model for performing the downstream tasks of probabilistic image reconstruction in the context of Bayesian inference of inverse problems for inpainting and denoising applications. Finally, we identify latent space density from NF as a promising outlier detection metric.

## 1 Introduction

Deep generative models are powerful machine learning models that can learn complex, high-dimensional data likelihoods and generate samples from them. Because of their probabilistic formulation, generative models are becoming an indispensable tool for scientific data analysis in a range of domains including particle physics (Paganini et al., 2018; Stein et al., 2020) and cosmology (Thorne et al., 2021; Reiman et al., 2020).

Variational Autoencoders (VAEs) (Kingma & Welling, 2014; Rezende et al., 2014) are among the most popular generative models. VAEs project the data to a lower dimensional latent space and reformulate the data likelihood estimation as a variational inference problem. Their training objective is the Evidence Lower BOund (ELBO), which approximates the true data likelihood with a variational ansatz from below. VAEs can be built with expressive architectures, enjoy the benefits of regularization through data compression and have a firm theoretical foundation. Different to generative adversarial networks (Goodfellow et al., 2014), another popular class of generative models, VAEs provide an estimator for the data likelihood and a posterior distribution for the latent variables.

Despite their popularity, variational autoencoders have well known practical limitations. Successful VAE training requires to find a delicate balance between the two contributing terms to the ELBO: The distortion term, which encourages high quality reconstructions, and the rate term, which controls the sample quality by matching the aggregate posterior with a chosen prior distribution (Alemi et al., 2018). Whether the VAE training process succeeds in striking this balance depends on a number of factors, including the network architectures, the chosen prior and the class of allowed posterior distributions (Hoffman & Johnson, 2016). In some cases, too powerful decoders can decouple the latent space from the input (Bowman et al., 2016; Chen et al., 2017) and lead to posterior collapse (van den Oord et al., 2017).

A long list of works have dissected and studied the training behavior of VAEs (Alemi et al., 2018; Hoffman & Johnson, 2016) and suggested modifications to remedy common issues. Many fixes add complexity to the VAE model, e.g. by modifying or annealing the ELBO objective (Bowman et al., 2016; Alemi et al., 2017; Higgins et al., 2017; Makhzani et al., 2015), choosing more expressive posterior distributions (Kingma et al., 2016; Rezende & Mohamed, 2015; Salimans et al., 2015; Tran et al., 2016), or using more flexible priors (Bauer & Mnih, 2019; Chen et al., 2017; Tomczak & Welling, 2018).

In this work we take a different approach. We give up on the variational ansatz that lies at the heart of VAEs and instead suggest a conceptually very simple model with very stable training properties. The Probabilistic Autoencoder (PAE) is motivated by probabilistic principal component analysis (Tipping & Bishop, 1999) and consists of an Autoencoder (AE), which is interpreted probabilistically after training by means of a Normalizing Flow (NF). Both of these components are comparably easy to set up and train and this two-stage set up allows the practitioner to optimize their hyper-parameters (model architecture, training procedure, etc.) independently. We claim that the PAE is a viable alternative to VAEs despite its conceptual simplicity. We back this claim empirically through ablation studies. Specifically, we compare the performance of the PAE to that of equivalent VAEs in a number of tasks which we think are specifically relevant for practical applications: data compression (reconstruction quality), data generation, anomaly detection and probabilistic data denoising and imputation.

Our primary contributions are: 1) a simple generative model designed with ease-of-use and training in mind 2) a quantitative comparison of this model to variational autoencoders, showing that it performs relevant tasks at comparable quality and accuracy without variational inference 3) a new anomaly detection metric through NF density estimation in latent space, which is a byproduct of the PAE, but can also be used within the VAE framework. We make all of our code publicly available.[1]

## 2 Motivation: Probabilistic PCA

The probabilistic autoencoder is motivated by linear Principal Component Analysis (PCA) and its probabilistic interpretation, probabilistic principal component analysis (Tipping & Bishop, 1999), which provides a PCA-based data likelihood estimate.

A principal component analysis of data $\boldsymbol{x} \in \mathbb{R}^N$ at fixed latent space dimensionality $K$ ($K<N$) finds the orthogonal linear transformation, $\boldsymbol{O}$,

$$\boldsymbol{O} : \mathbb{R}^K \to \mathbb{R}^N, \boldsymbol{z} \mapsto \boldsymbol{Oz}, \boldsymbol{OO}^T = \mathbb{1}_N \tag{1}$$

that maximizes the data variance in the latent space. Maximizing the variance of the transformed data is equivalent to minimizing the average reconstruction error (the residual variance in data space). The PCA problem can be solved analytically and the principal components are given by the eigenvectors of the data covariance matrix.

A suitable latent space dimensionality, $K$, is chosen by inspecting the eigenvalues, $\lambda_i$, of the data covariance and keeping only the eigenvectors that correspond to the largest eigenvalues. The average reconstruction error that originates from the discarded eigenvalues is $\sigma^2_{\text{recon}} = \sum_{i=K+1}^N \lambda_i$.

The data model under a PCA is

$$\boldsymbol{x} = \boldsymbol{Oz} + \boldsymbol{\epsilon}, \tag{2}$$

---

[1]https://github.com/VMBoehm/PAE-ablation

where $\boldsymbol{O}$ is constructed from the eigenvectors that correspond to the largest eigenvectors and $\boldsymbol{\epsilon}$ is the residual not captured by the PCA transformation.

In probabilistic PCA (PPCA) the residuals are assumed to follow a Gaussian distribution. The implicit likelihood is then,

$$\ln \tilde{p}(\boldsymbol{x}|\boldsymbol{z}) = -\frac{1}{2}\left[N\ln(2\pi) + \ln\det\boldsymbol{\Sigma} + (\boldsymbol{x} - \boldsymbol{O}\boldsymbol{z})^T\boldsymbol{\Sigma}^{-1}(\boldsymbol{x} - \boldsymbol{O}\boldsymbol{z})\right]. \tag{3}$$

Under the approximation that the reconstruction error is uncorrelated and isotropic, $\boldsymbol{\Sigma}$ is a diagonal matrix with $\sigma_{\text{recon}}^2$ along its diagonal, $\boldsymbol{\Sigma} = \sigma_{\text{recon}}^2 \mathbb{1}_N$.

The implicit likelihood in equation 3 alone is not yet a probabilistic model for the data. A fully probabilistic structure requires a prior over the latent space. PPCA (Tipping & Bishop, 1999) assumes that the latent variables follow a Gaussian distribution with mean zero and covariance $\boldsymbol{\Lambda}$, where $\boldsymbol{\Lambda}$ is a diagonal matrix with the rank-ordered eigenvalues $\lambda_i$ along its diagonal.

The Gaussian prior allows us to analytically compute the marginal,

$$\ln \tilde{p}(\boldsymbol{x}) = -\frac{1}{2}\left[N\ln(2\pi) + \ln\det\boldsymbol{C} + \boldsymbol{x}^T\boldsymbol{C}^{-1}\boldsymbol{x}\right], \tag{4}$$

with $\boldsymbol{C} = \boldsymbol{O}\boldsymbol{\Lambda}\boldsymbol{O}^T + \boldsymbol{\Sigma}$.

In summary, probabilistic PCA constructs a probabilistic model by first finding a basis which minimizes the reconstruction error in a class of models, followed by using the probability distribution of the latent variables as the prior. With the probabilistic autoencoder we generalize this approach to non-linear models. The PCA is replaced by an autoencoder trained to minimize the reconstruction error, and the Gaussian ansatz for the prior is replaced by a normalizing flow.

## 3 The probabilistic autoencoder

### 3.1 PAE training

In analogy to PPCA the PAE is constructed in two stages. Stage 1 is an autoencoder with encoder $\boldsymbol{f}$ and decoder $\boldsymbol{g}$, both deep neural networks with respective trainable parameters $\phi$ and $\theta$,

$$\boldsymbol{f}_\phi : \mathbb{R}^N \to \mathbb{R}^K, \boldsymbol{x} \mapsto \boldsymbol{f}_\phi(\boldsymbol{x}), \ \boldsymbol{g}_\theta : \mathbb{R}^K \to \mathbb{R}^N, \boldsymbol{z} \mapsto \boldsymbol{g}_\theta(\boldsymbol{z}). \tag{5}$$

The training objective of the AE is the reconstruction error or $L_2$-distance,

$$\mathcal{L}_{\text{AE}} = \mathbb{E}_{p(\boldsymbol{x})}\,||\boldsymbol{x} - \boldsymbol{g}_\theta(\boldsymbol{f}_\phi(\boldsymbol{x}))||_2^2. \tag{6}$$

The autoencoder is not a probabilistic model. To construct the PAE, we interpret it probabilistically with a second stage. We approximate the latent space prior, $p(\boldsymbol{z})$, by performing a density estimation on the AE-encoded training data. In PPCA the latent space density is modeled with a Gaussian. The PAE employs a more flexible density estimator for modeling the prior, a normalizing flow.

Normalizing flows (Rippel & Adams, 2013; Dinh et al., 2015; 2017; Kingma & Dhariwal, 2018; Grathwohl et al., 2019) address the task of modeling the density distribution $p(\boldsymbol{z})$ of input data $\boldsymbol{z}$ by introducing a bijective mapping, $\boldsymbol{b}_\gamma(\boldsymbol{z}) = \boldsymbol{u}$, from the data $\boldsymbol{z}$ to an underlying latent representation $\boldsymbol{u}$,

$$\boldsymbol{b}_\gamma : \mathbb{R}^K \to \mathbb{R}^K, \boldsymbol{z} \mapsto \boldsymbol{u} = \boldsymbol{b}_\gamma(\boldsymbol{z}). \tag{7}$$

Requiring the latent variables to follow a given prior distribution $q(\boldsymbol{u})$, one can write the modeled data probability density using conservation of probability,

$$p_\gamma(\boldsymbol{z}) = q(\boldsymbol{u})|\nabla_{\boldsymbol{z}}\boldsymbol{b}_\gamma(\boldsymbol{z})|. \tag{8}$$

Here, $q(\boldsymbol{u})$ is some simple normalized latent space probability density, usually a Gaussian $\mathcal{N}(\boldsymbol{0}, \boldsymbol{I})$, and $|\nabla_{\boldsymbol{z}}\boldsymbol{b}_\gamma(\boldsymbol{z})|$ is the Jacobian determinant of the transformation $\boldsymbol{b}_\gamma(\boldsymbol{z})$. The NF is parametrized by parameters

$\gamma$ and training takes the form of maximizing the data likelihood $p_\gamma(\boldsymbol{z})$ with respect to $\gamma$. Equation 8 requires an evaluation of the Jacobian. NF architectures have forms for which this is simple and fast. The architectural constraints originating from this requirement also introduce beneficial regularizing properties and prevent overfitting. A schematic diagram of the PAE is shown in figure 1.

In the PAE, an NF is trained as a mapping from the latent space of the AE, $\boldsymbol{z}$, to the Gaussian latent space of the normalizing flow, $\boldsymbol{u}$. The training objective of the NF is the negative log likelihood of the encoded samples, $\boldsymbol{z} = f_\phi(\boldsymbol{x})$,

$$\mathcal{L}_{\mathrm{NF}} = \mathbb{E}_{\tilde{p}(\boldsymbol{z})}[-\ln p_\gamma(\boldsymbol{z})] = \mathbb{E}_{\tilde{p}(\boldsymbol{z})} \left[ -\ln p(\boldsymbol{u}) - \ln \left| \det \frac{\partial \boldsymbol{b}_\gamma^{-1}(\boldsymbol{u})}{\partial \boldsymbol{u}} \right| \right]_{\boldsymbol{u} = \boldsymbol{b}_\gamma(\boldsymbol{z})}. \tag{9}$$

The normalizing flow maps the potentially very irregular latent space distribution of the AE to a Gaussian distribution.

To sample from the PAE we draw a sample, $\boldsymbol{u} \sim \mathcal{N}(\boldsymbol{0}, \boldsymbol{1})$, from the NF latent distribution and pass it through both the NF and AE generators (left panel in figure 1),

$$\boldsymbol{x} = \boldsymbol{g}_\theta(\boldsymbol{b}_\gamma^{-1}(\boldsymbol{u})). \tag{10}$$

Just as in PPCA, density estimation on the encoded data only provides an approximate prior. Formally, for a fully probabilistic model, a prior would be given by the aggregate posterior,

$$p_{\mathrm{model}}(\boldsymbol{z}) = \int \mathrm{d}\boldsymbol{x}\, p(\boldsymbol{x})\, p_{\mathrm{model}}(\boldsymbol{z}|\boldsymbol{x}) = \mathbb{E}_{p(\boldsymbol{x})} \left[ p_{\mathrm{model}}(\boldsymbol{z}|\boldsymbol{x}) \right], \tag{11}$$

meaning that the density estimation should be conducted on samples from the posteriors. Since our first stage is a non-probabilistic autoencoder, there is no notion of a posterior. Our choice to fit an approximate prior on the encoded data, however, is not completely unjustified: we know that small reconstruction errors (which can be easily achieved with an AE) generally result in very narrow posteriors centered on latent space points which are well determined through the projection of the data into the latent space. The position of these points are hardly influenced by the prior. By replacing samples from the posterior with the encoded samples, we replace the narrow posteriors by Dirac delta distributions. By fitting on the AE encoded data we approximate the maximum a posteriori (MAP) solution with the AE encoded position. This is not a formal mathematical derivation, but can serve as a reasoning for why the PAE is able to compete with fully probabilistic models even in probabilistic tasks.

The generalization and regularization properties of NFs are another important ingredient for enabling the use of approximate MAP positions instead of samples from the posterior. NFs are unlikely to fit delta functions to their training points, but smoothly interpolate between them. The success of NF-based density estimation is proof of this: on many datasets NFs achieve the highest validation log data likelihoods (Durkan et al., 2019). If the NF does not provide sufficient regularization, it will become apparent as overfitting (lower density estimates on validation data), which can be controlled by simplifying the architecture or early stopping.

The AE latent space is usually of relatively low dimensionality, $K \ll N$, which allows for computationally tractable density estimation: the NF models do not require complex deep architectures and are fast to train, which enables efficient hyper-parameter optimization.

## 3.2 Comparison to VAE

Different to the PAE a variational autoencoder is trained on a fully probabilistic objective, the Evidence Lower BOund (ELBO),

$$\mathrm{ELBO} = -\mathcal{L}_{\mathrm{VAE}} = \mathbb{E}_{p(\boldsymbol{x})} \left[ \mathbb{E}_{q_\phi(\boldsymbol{z}|\boldsymbol{x})} \left[ \ln p_\theta(\boldsymbol{x}|\boldsymbol{z}) \right] - \mathrm{D}_{\mathrm{KL}} \left[ q_\phi(\boldsymbol{z}|\boldsymbol{x}) || p(\boldsymbol{z}) \right] \right], \tag{12}$$

where $q_\phi$ is an approximate, parametrized variational posterior, usually a Gaussian with diagonal covariance. Typical choices for the parametrized implicit likelihood $p_\theta(\boldsymbol{x}|\boldsymbol{z})$ are a Bernoulli distribution for binary valued

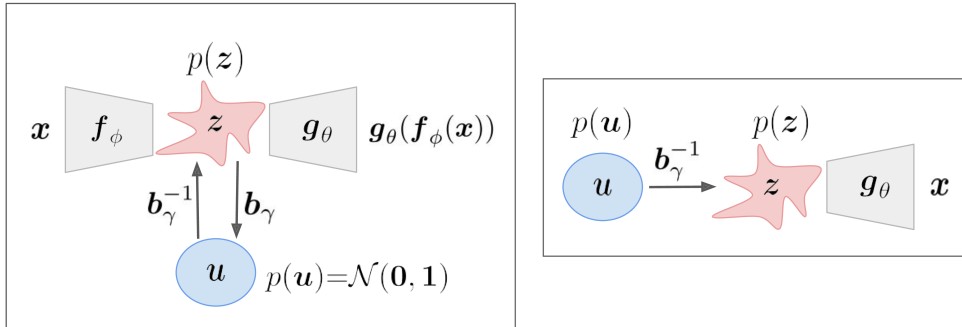

Figure 1: Schematic diagram of the PAE (left panel) and an illustration of the sampling procedure from the PAE (right panel). The autoencoder networks are depicted as gray trapezia, the normalizing flow is represented by black arrows and the latent spaces of the autoencoder and normalizing flow are shown in red and blue, respectively.

data or a diagonal Gaussian distribution for continuous data. The ELBO is guaranteed to bound the true evidence $p(\boldsymbol{x})$ from below. During the VAE training equation 12 is evaluated stochastically on samples from the approximate posterior. Equation 12 shows that the VAE objective balances the average reconstruction error (the likelihood term or distortion) with the sample quality (the KL term or rate). If the former dominates the loss during training, the encoded distribution and prior do not match well. If this is the case, samples from the prior can land outside of the encoded domain resulting in low sample quality. If the KL term dominates, some latent dimensions will solely be used to satisfy the second term and not encode any information about the input data, a problem known as posterior collapse (Alemi et al., 2018). Balancing the two terms can be controlled by an additional parameters $\beta$,

$$\mathcal{L}_{\beta-\text{VAE}} = - \mathbb{E}_{p(\boldsymbol{x})} \left[ \mathbb{E}_{q_\phi(\boldsymbol{z}|\boldsymbol{x})} \left[ \ln p_\theta(\boldsymbol{x}|\boldsymbol{z}) \right] - \beta \, \mathrm{D}_{\text{KL}} \left[ q_\phi(\boldsymbol{z}|\boldsymbol{x}) || p(\boldsymbol{z}) \right] \right]. \tag{13}$$

This and related modification are known as $\beta$-VAEs (Bowman et al., 2016; Alemi et al., 2017; Higgins et al., 2017; Makhzani et al., 2015). Training on equation 13 usually involves a grid search in order to find an optimal value for $\beta$ and annealing schedules.

The PAE optimizes the reconstruction and sample quality individually. Training of stage 1 reaches an optimal reconstruction error. The latter is then left unchanged in the training of stage 2, which can focus entirely on matching the latent space distribution. We test in our experiments whether this procedure results in an advantage in reconstruction error and sample quality. A practical advantage of this procedure is that it facilitates the hyper-parameter search over model architecture and training schedule. Instead of having to iterate over encoder/decoder and flow architecture and the balance between rate and distortion term, each step can be optimized individually and towards a single objective.

For our comparisons between VAE and PAE to be fair, we allow the VAE prior, which is typically a standard normal distribution, to be more flexible. In analogy to the PAE, we model it with a normalizing flow,

$$\mathcal{L}_{\text{flow}-\text{VAE}} = - \mathbb{E}_{p(\boldsymbol{x})} \left[ \mathbb{E}_{q_\phi(\boldsymbol{z}|\boldsymbol{x})} \left[ \ln p_\theta(\boldsymbol{x}|\boldsymbol{z}) \right] - \mathrm{D}_{\text{KL}} \left[ q_\phi(\boldsymbol{z}|\boldsymbol{x}) || p_\gamma(\boldsymbol{z}) \right] \right]. \tag{14}$$

## 4 Downstream Tasks

In our experiments, we test the PAE performance not only in terms of sample and reconstruction quality, but also in terms of anomaly detection, a highly relevant downstream task of generative models. In appendix E, we further show how the PAE can be used for posterior-based probabilistic image inputation.

### 4.1 Anomaly detection

One application of generative models is anomaly or out-of-distribution (OoD) detection. This is often based on the assumption that a density estimator should return smaller probability densities for out-of-distribution

data than in-distribution (iD) data. However, this is assumption is not always satisfied and generative model based density estimators have been reported to perform poorly in some OoD detection problems. OoD detection with VAEs, NFs (Kingma & Dhariwal, 2018) and PixelCNNs (van den Oord et al., 2016) can exhibit catastrophic outlier detection failures (Nalisnick et al., 2019).

The PAE model is not trained to maximize the data likelihood, nor does it provide an estimate of it. While we could attempt to perform a marginalization over the latent space (after introducing an approximate implicit likelihood) in order to obtain such an estimate, we suggest a much simpler outlier detection metric: the estimated density in latent space. We find in our experiments that the NF estimated latent space density is an excellent OoD detection metric for outlier detection problems that have been identified as problematic in the literature.

Our OoD metric is again motivated by PPCA: The PPCA model is a useful toy model to understand how dimensionality reduction prior to density estimation can be beneficial. The PPCA density estimate in latent space is given by

$$\ln \tilde{p}(\boldsymbol{z}) = -\frac{1}{2} \left[ K \ln(2\pi) + \sum_i^K \ln \lambda_i + \sum_i^K z_i \lambda_i^{-1} z_i \right]. \tag{15}$$

Equation 15 diverges for $\lambda_i \to 0$. This suggests that $\tilde{p}(\boldsymbol{z})$ and henceforth $\tilde{p}(\boldsymbol{x})$ (Equation 4) can be dominated by small eigenvalues. These small eigenvalue components are well known to be difficult to estimate from a limited amount of data, i.e. they are prone to overfitting. This has led to the development of special co-variance estimation regularization techniques known as shrinkage methods in the statistics literature (Chen et al., 2010; Ledoit & Wolf, 2004). Dimensionality reduction keeping the largest eigenvalues removes dimensions with vanishing variance and hence cures the estimator's sensitivity to small and likely mis-estimated eigenvalues. In PPCA $\tilde{p}(\boldsymbol{x})$, the data covariance $\boldsymbol{C} = \boldsymbol{O}^T \boldsymbol{\Lambda} \boldsymbol{O} + \boldsymbol{\Sigma}$ is regularized by the noise covariance, $\boldsymbol{\Sigma}$, which reinterprets the discarded and mis-estimated eigenvalues as noise. Noise is not informative for anomaly detection, which suggests that PPCA data space density estimation has no advantage over density estimation in latent space for this task.

We illustrate this on an outlier detection problem between the FashionMNIST (Xiao et al., 2017) and MNIST (Lecun et al., 1998) data sets in figure 2. Both of these data sets have a data covariance matrix with a high condition number. Only 86 PCA components are required to capture 90% of the data variance in MNIST (compared to $N=784$) and the smallest eigenvalues are evidently singular. Similar applies to F-MNIST, where a PCA captures 90% of the data variance with 83 components. These data are known to produce catastrophic failures in OoD detection, specifically when presenting samples from the MNIST data set to models trained on F-MNIST (Nalisnick et al., 2019). We construct a probabilistic PCA model for F-MNIST (from training data) and use equation 4 and equation 15 as outlier detection metrics to separate F-MNIST in-Distribution (iD) test data from MNIST Out-of-Distribution (OoD) data. In figure 2 we show the Area Under Receiver Operator Curve for this outlier detection task as a function of the number of PCA components. The highest outlier detection accuracy based on latent space density estimation (equation 15) is AUROC = 0.980 and it is reached at a relatively low number of PCA components of 127. The highest accuracy for OoD detection based on data space density estimation (equation 4) is AUROC = 0.974 at 37 components. We argue analogously that the PAE latent space density is better for OoD detection than using a full dimensionality NF. In our experiments, we find that the PAE latent space density is a superior anomaly detector than the ELBO or full dimensionality NF.

## 5 Related Work

Generative moment matching networks (Li et al., 2015) have been proposed to be used in a 2-stage PAE-like set up, consisting of an autoencoder and a mapping of a Gaussian to the encoded distribution. The second stage is non-invertible and trained with a moment-matching objective. Wasserstein autoencoders (Tolstikhin et al., 2018) employ a training objective based on the Wasserstein distance to match the encoded distribution to a given prior. WAEs achieve high sample quality, but do not provide a density estimate. Generative latent flows (Xiao et al., 2019) are a similar conjunction of AE and NF and achieve high sample quality, but the

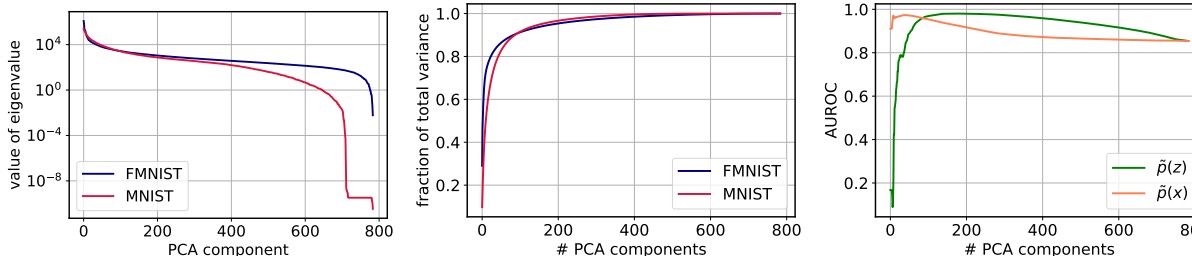

Figure 2: Principal component analysis of FashionMNIST and MNIST data sets and outlier detection accuracy (in-distribution: Fashion MNIST, out-of-distribution: MNIST) with equation 4 and equation 15 as a function of included number of PCA components. A higher AUROC value corresponds to a better separation between in- and out-of-distribution data.

authors do not perform controlled ablation studies to compare their approach with ELBO-based training objectives, nor do they explore the probabilistic interpretation for downstream tasks.

Giving more flexibility to the prior distribution has also been suggested in the context of VAEs, e.g. by modeling it with an NF (Bauer & Mnih, 2019; Chen et al., 2017; Tomczak & Welling, 2018). This has sometimes been deemed prone to overfitting (Tomczak & Welling, 2018) and many works suggest using more expressive variational distributions instead (Kingma et al., 2016; Rezende & Mohamed, 2015; Salimans et al., 2015; Tran et al., 2016). In our experiments, we compare our PAE with a VAE with an NF prior because it allows us to compare models with the same architecture. We pay special attention to overfitting, but do not observe it to be a problem.

Other approaches that improve the VAE sample quality include $\beta$-VAEs (Tishby et al., 2000; Alemi et al., 2017; Hledík et al., 2019; Higgins et al., 2017) and 2-Stage-VAEs (Dai & Wipf, 2019). $\beta$-VAEs balance rate and distortion by means of an additional scalar parameters ($\beta$). 2-stage-VAEs combine two consecutive VAE's, one for the purpose of data compression, where the KL term in the ELBO is suppressed and a second one for latent space density estimation. This two-stage approach achieves high quality samples. With the PAE we demonstrate that this success does not rely on the ELBO objective in the first stage and that the first stage can be replaced by an AE and the second stage by a powerful NF density estimator. High sample quality is also achieved by VQ-VAE (van den Oord et al., 2017; Razavi et al., 2019), another model that requires 2-stage training. It combines a more complicated first stage which includes hyper-parameter tuning and discretization with a second stage in which an autoregressive model is trained to learn the prior. The successes of 2-stage models indicate that separating the tasks of learning a lower-dimensional representation and learning its distribution is beneficial for generative model performance.

Other non-ELBO approaches that address density estimation for data that is confined to a lower dimensional manifold include $\mathcal{M}_{(e)}$-flows (Brehmer & Cranmer, 2020) and relaxed injective probability flows (Kumar et al., 2020). Instead of separating the tasks of compression and density estimation in the lower dimensional manifold, these models regularize a flow itself. The authors of $\mathcal{M}_{(e)}$-flows compare their model with the PAE and find that the PAE reaches the same low reconstruction error and a higher accuracy in anomaly detection. The recently introduced regularized autoencoder is another deterministic VAE alternative based on an autoencoder, that produces comparable or even better sample quality than VAEs (Ghosh et al., 2020).

Downstream tasks: Out-of-distribution detection with generative models has recently attracted a lot of attention, triggered by the finding that state-of-the art generative models such as VAE, GLOW (Kingma & Dhariwal, 2018) and MAF (Papamakarios et al., 2017) fail in this task on a number of standard data sets (Nalisnick et al., 2019). Our finding that dimensionality reduction combined with latent space density estimation results in reliable anomaly detection is in line with other works: e.g. Abati et al. (2019) use a single stage training with a free parameter that controls the relative contribution of reconstruction error and NF latent space density and requires tuning of the free parameter. They apply it to OoD, but not to other PAE tasks such as generating samples, denoising and inpainting. Other proposed solutions include the use likelihood ratios as an OoD metric instead of the likelihood itself (Ghosh et al., 2020). More recently,

a reliable OoD detection was reported with density-of-states, a method which leverages another density estimator on top of the density estimation (Morningstar et al., 2021).

# 6 Experiments

The aim of our experiments is to test whether training on the ELBO offers any measurable advantage over the PPCA-inspired PAE training approach. For this test, we construct and train equivalent PAE and VAE models and compare them in terms of reconstruction error, sample quality, outlier detection accuracy and probabilistic inpainting and denoising ability. We conduct our detailed comparisons between PAE and VAE models on the FashionMNIST data set. For anomaly detection we perform an additional comparison on MNIST and CIFAR10 with the anomaly detection method by Abati et al. (2019) (Appendix C). We further train a PAE model on the higher dimensional Celeb-A (Liu et al., 2015) data set (Appendix A).

## 6.1 Ablation studies

We compare the PAE to two ELBO-based alternatives:

1. A VAE with a normalizing flow prior, where the encoder/decoder pair is trained on equation 13 with $\beta=0$, i.e. without the KL-Divergence term and the variance of $q(z)$ is kept constant. The normalizing flow is trained in a second stage on the encoded distribution. The ELBO with $\beta=1$ and the normalizing flow prior is used as a density estimator. We call this model $\boldsymbol{\beta_0}$-**VAE**. The difference to the PAE lies in the noisy estimation of the likelihood during training. Comparing the PAE to the $\boldsymbol{\beta_0}$-VAE tests whether training on the reconstruction error instead of the distortion term offers any advantage.

2. A VAE with a normalizing flow prior that is trained on the ELBO with $\beta=1$. We call this model **flow-VAE**. The difference to the PAE lies in the training procedure. In the flow-VAE, the normalizing flow, encoder and decoder are trained jointly under the ELBO training objective. This means that the normalizing flow is trained on samples from the approximate posterior instead of the encoded samples, which are used in the PAE.

Table 1 summarizes the different models and the parameters in which they differ. To allow for a fair comparison, we use the same model architecture and training parameters for all experiments, except in cases where fixing them might disadvantage the VAE. Encoder and decoder networks are loosely based on the infoGAN architecture (Chen et al., 2016). Normalizing flows are constructed from RealNVP transformations (Dinh et al., 2017), Neural Spline Flow (NSF) transformations (Durkan et al., 2019) and trainable permutations (Kingma & Dhariwal, 2018) (GLOW). The exact model architectures are listed in table F and the choice of parameters is detailed below in table 6.3. Because we use non-binarized data, we use a Gaussian implicit likelihood in our VAE models (instead of a Bernoulli likelihood which would be the suitable choice for binarized data). The value of the scale parameter, $\sigma$, of this implicit likelihood was set to $\sigma = 0.1$. We found this to be to be the optimal value in a small ablation study (appendix G). We also found that the model performance is not overly sensitive to this choice. In addition to the experiments presented here, we trained a vanilla $\beta$-VAE and show results in appendix B. We did not include this model in main text because it does not use a flow prior and requires parameter fine-tuning. We also find that it performs worse than all alternatives studied in the main text.

## 6.2 Data sets and preprocessing

We perform our ablation studies on the Fashion-MNIST (Xiao et al., 2017) data set, which we split into 50,000 training examples, 10,000 validation and 10,000 test samples. As outlier data sets we use MNIST (Lecun et al., 1998) and Omniglot (Lake et al., 2015), as well as horizontal and vertical flips of F-MNIST test data. We preprocess the data by dequantizing (adding uniform noise $\in [-1/256, 1/256]$) before rescaling pixel values to the interval [-0.5,0.5].

| model parameters | | | |
|---|---|---|---|
| **model name** | $\beta_0$-VAE | PAE | flow-VAE |
| training objective(s) | $\mathcal{L}_{\beta-\text{VAE}}$, $\mathcal{L}_{\text{NF}}$ | $\mathcal{L}_{\text{AE}}$,$\mathcal{L}_{\text{NF}}$ | $\mathcal{L}_{\text{flow}-\text{VAE}}$ |
| flow prior | TRUE | TRUE | TRUE |
| 2 stage training | TRUE | TRUE | FALSE |
| $\beta$ | 0 | N/A | 1 |
| dropout rate | 0.15 | 0.15 | N/A |
| OoD metric | ELBO | $\log p(z)$ | ELBO |

Table 1: Overview of the different models used in the ablation studies.

## 6.3 Parameter choice and training procedure

**Training of the encoder/decoder pair:** We used the same encoder and decoder architecture for all of our experiments. We further fixed the latent space dimensionality to 40, the number of training steps to 300,000 and used a learning rate schedule in which we keep the learning rate constant at the initial value up to training step 100,000, then reduce it linearly down to 1/10 of the initial rate over 50,000 steps. For the remaining 150,000 steps we restart the learning rate and repeat the annealing scheme. We did not find our final loss to depend on the details of this annealing scheme, but found that annealing and restarting was beneficial. We used the ADAM optimizer (Kingma & Ba, 2015) with parameters $\beta_1 = 0.9, \beta_2 = 0.999, \epsilon = 10^{-7}$ in all of our trainings (including for the normalizing flow).

The batch size, initial learning rate, sample size in the stochastic evaluation of the ELBO as well as the drop out rate were optimized to yield the best possible reconstruction error on validation data on the $\beta_0$-VAE model: we ran around 30 shorter trainings (100,000 steps) with different combinations of these parameters and used a Gaussian Process surrogate to determine the combination of parameter values that optimized the reconstruction error on the validation data. The values we obtained through this procedure are outlined in table 2 and were used in all experiments. The only parameter that we adapted in some of our experiments is the dropout rate. The dropout layer is a necessary regularization to prevent overfitting in autoencoder models and models with $\beta = 0$. This regularization is not necessary in VAE models with a flow prior and was not used in these models. We chose the $\beta_0$-VAE for parameter optimization, because we think that it is a fair middle ground between PAE and flow-VAE sharing some properties with both. It was also chosen for convenience, because it allowed us to optimize the encoder/decoder pair and and normalizing flow separately, without having to worry about the feedbacks of changing one on the other. We point out that optimizing on the flow-VAE would have suffered from this complication. To not disadvantage the flow-VAE, we also train a flow-VAE with a different (simpler) NF architecture.

| common model parameters | |
|---|---|
| batch size | 256 |
| initial learning rate | 1.23E-03 |
| learning rate annealing | TRUE |
| sample size | 16 |
| training steps | 300000 |
| latent size | 40 |

Table 2: Training parameters that were kept constant in all encoder/decoder pair trainings. The parameters were optimized to achieve the lowest reconstruction error on validation data for the $\beta_0$-VAE model.

**Training of normalizing flows:** We construct normalizing flows from realNVP, NSF and GLOW building blocks. We optimized the number of building blocks of each kind to achieve maximal $\log p_\gamma(z)$ on the validation data encoded with the $\beta_0$-VAE model. In models where the NF was trained separately, we used a learning schedule consisting of 120 epochs during which both learning rate and batch size were annealed and restarted. We found that a restarting learning rate was especially helpful when using NSF transformation

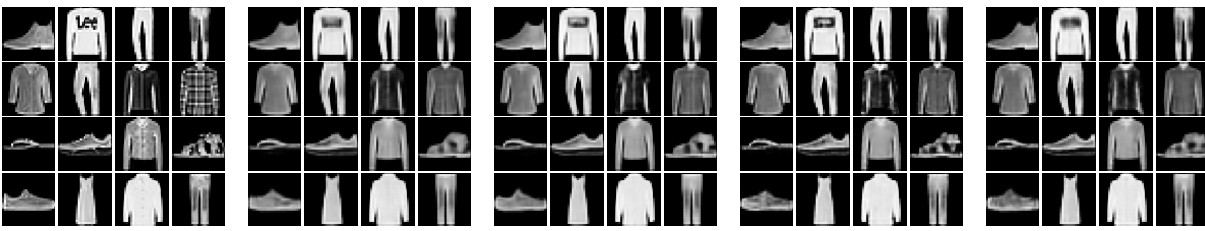

Figure 3: Visual comparison of F-MNIST reconstructions with different models. From left to right: original data, $\beta_0$-VAE, PAE, flow-VAE, flow-VAE(s).

layers, but that the final loss did not strongly depend on the details of the annealing and restarting scheme. **Training of VAE with normalizing flow prior:** In the flow-VAEs, encoder, decoder and NF are trained jointly on the ELBO objective with the same training parameters and architectures that were used for $\beta_0$-VAE and PAE. The only parameter that we adapted was the dropout rate, which we set to zero. To test the dependence on the flow architecture, we ran two experiments. One in which we adapted the flow architecture we had found to work best on the $\beta_0$-VAE encoded data, and another experiment, in which we simplified the flow architecture (flow-VAE (s)). We found the deeper architecture to be superior. All NF architectures are detailed in appendix F.

**Robustness of results:** For every model (PAE, $\beta_0$-VAE, flow-VAE) we repeated every training run three times starting from different initial network parameter values. We report results for the trained models that achieved the lowest training loss on validation data out of these three runs. We found very little scatter between different trainings of the same model.

### 6.4 Results

**Reconstruction Quality:** We measure the reconstruction quality in all models by means of the average reconstruction error on the test data set,

$$\overline{\sigma_{\text{recon}}^2} = \frac{1}{M}\frac{1}{N}\sum_j^M\sum_i^N\left[x_{ji} - \boldsymbol{g}_\theta(\boldsymbol{f}_\phi(\boldsymbol{x}_j))_i\right]^2 = \frac{1}{M}\sum_j^M \sigma_{\text{recon},j}^2, \tag{16}$$

where $j$ labels the image and $i$ the pixel. In the VAE models we evaluate the reconstruction error using the mean of the variational posterior. We also report the 95 percentile of the image-wise reconstruction errors $P_{95\%}(\sigma_{\text{recon}}^2)$, which better characterizes the large error tail of the distribution. The results are listed in table 3, the reported errors were obtained through bootstrapping on the test data.

| | reconstruction quality | | | |
|---|---|---|---|---|
| **model name** | $\beta_0$-VAE | PAE | flow-VAE | flow-VAE(s) |
| $\sigma_{\text{recon}}^2\ [\times 10^{-3}]\ (\downarrow)$ | $6.24 \pm 0.06$ | $\mathbf{5.87\pm0.06}$ | $6.22 \pm 0.07$ | $6.66 \pm 0.06$ |
| $P_{95\%}(\sigma_{\text{recon}}^2)\ [\times 10^{-3}]\ (\downarrow)$ | $29.6 \pm 0.3$ | $\mathbf{27.6 \pm 0.3}$ | $35.7 \pm 0.4$ | $31.9 \pm 0.3$ |

Table 3: Comparison of F-MNIST models in terms of reconstruction quality. The PAE model achieves the lowest reconstruction errors.

We find that the PAE model has a consistently and significantly lower mean and 95 percentile reconstruction error than ELBO-based models. In figure 3, we show reconstructions of test data points for each model.

**Sample Quality:** How to best measure image generation quality in a way that quantifies both image quality (are the samples visually compelling?) and diversity (is the model sampling from the full distribution?), is still a question of active research. Here, we measure the sample quality in terms of the well established Frechet-Inception Distance (FID) score (Heusel et al., 2017), which is known to correlate well with human perception of image quality. The FID score takes a sample of generated images and a sample of real

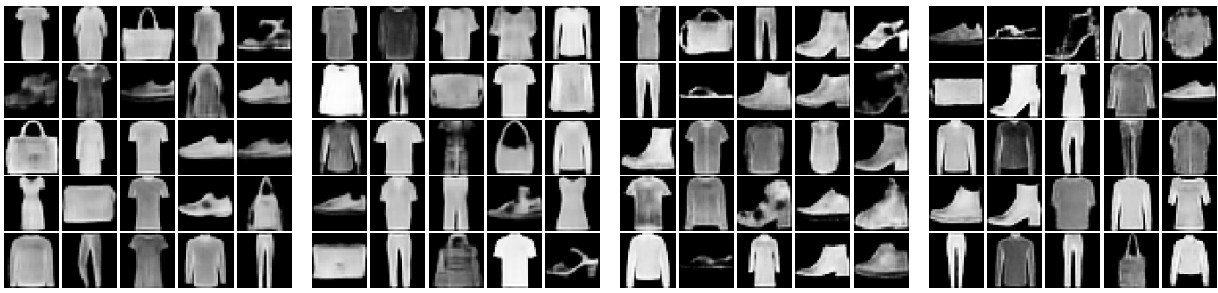

Figure 4: Visual comparison of random samples. From left to right: $\beta_0$-VAE, PAE, flow-VAE, flow-VAE (s).

images, passes each of them through the pre-trained Inception network (Szegedy et al., 2016) and extracts as features the outputs of one of the last layers. Each set of features is then fitted with a multivariate Gaussian distribution, $\mathcal{N}(x|\mu, \Sigma)$. The FID score is defined as the Fréchet distance between these two Gaussians

$$\text{FID}(x_{\text{orig}}, x_{\text{gen}}) = ||\mu_{\text{orig}} - \mu_{\text{gen}}||_2^2 + \text{Tr}\left(\Sigma_{\text{orig}} + \Sigma_{\text{gen}} - 2(\Sigma_{\text{orig}}\Sigma_{\text{gen}})^{\frac{1}{2}}\right). \tag{17}$$

In figure 4 we show generated images from each model. While visually comparable, we get the lowest FID scores for the flow-VAE model (table 4). A concern sometimes raised with models that use flow priors is

**sample quality**

| model name | $\beta_0$-VAE | PAE | flow-VAE | flow-VAE (s) |
|---|---|---|---|---|
| FID score ($\downarrow$) | $32.8 \pm 0.4$ | $28.4 \pm 0.3$ | $\mathbf{22.8 \pm 0.3}$ | $33.9 \pm 0.4$ |

Table 4: Sample quality, measured in terms of the FID score. Lower values are better. Error estimates obtained through bootstrapping.

that the added expressiveness of the model results in overfitting or memorization of training data. We pay special attention to overfitting during training by monitoring the loss on validation data.

**Outlier Detection Accuracy:** For outlier detection, we use the outlier detection metrics listed in table 1 and report the outlier detection accuracy for each model in terms of the Area Under Receiver Operator Curve (AUROC). An AUROC of 1 corresponds to a perfect separation between the out-of-distribution and in-distribution data. For the latter we use the F-MNIST test data set. The ELBO of the VAE is evaluated using 10 samples from the variational posterior and we verified that using more samples does not achieve better AUROC values. The results are listed in 5. The latent space density estimation of the PAE model outperforms the ELBO of our VAE models in all but one of our experiments. However, we do not observe a catastrophic failure of OoD detection with the ELBO as it has been reported for this data set (Nalisnick et al., 2019). Following our discussion in 4.1, which suggests that latent space density estimation is superior

**OoD detection accuracy measured in terms of AUROC $\uparrow$**

| OoD data | $\beta_0$-VAE | PAE (AE+flow) | flow-VAE | flow-VAE (s) |
|---|---|---|---|---|
| MNIST | $0.9086 \pm 0.0023$ | $\mathbf{0.9970 \pm 0.0003}$ | $0.9633 \pm 0.0014$ | $0.9230 \pm 0.0022$ |
| Omniglot | $0.8668 \pm 0.0025$ | $\mathbf{0.9736 \pm 0.0011}$ | $0.9221 \pm 0.0020$ | $0.8753 \pm 0.0021$ |
| F-MNIST hor. | $0.6790 \pm 0.0036$ | $\mathbf{0.6883 \pm 0.0038}$ | $0.6758 \pm 0.0036$ | $0.6773 \pm 0.0035$ |
| F-MNIST vert. | $0.8891 \pm 0.0022$ | $0.8789 \pm 0.0024$ | $\mathbf{0.8994 \pm 0.0022}$ | $0.8964 \pm 0.0019$ |

Table 5: Out of distribution detection with different models. The outlier detection accuracy is measured in terms of the AUROC$\in [0, 1]$. Higher values are better. Error estimates obtained through bootstrapping.

to data space density estimation in many cases, we disseminate the ELBO further and take a look at the different contributing terms of the ELBO. Specifically, we split the ELBO into three terms, which we identify

as distortion (first term), entropy (second term) and cross entropy (last term),

$$\text{ELBO} = \mathbb{E}_{p(\boldsymbol{x})}\left[\mathbb{E}_{q_\phi(\boldsymbol{z}|\boldsymbol{x})}\left[\ln p_\theta(\boldsymbol{x}|\boldsymbol{z}) - q_\phi(\boldsymbol{z}|\boldsymbol{x}) + p_\gamma(\boldsymbol{z})\right]\right], \tag{18}$$

and measure the outlier detection accuracy with each of these terms individually. The results are listed in table 6.

We find that the cross entropy term, which measures a stochastic mean over the latent space density and is therefore closely related to the latent space density, is a very reliable outlier detection metric for models trained on the ELBO objective.

**OoD detection with flow-VAE, AUROC($\uparrow$)**

| OoD data | distortion | rate | entropy | cross entropy |
|---|---|---|---|---|
| MNIST | $0.9620 \pm 0.0014$ | $\mathbf{0.9965 \pm 0.0003}$ | $0.9659 \pm 0.0012$ | $\mathbf{0.9963 \pm 0.0004}$ |
| Omniglot | $0.9176 \pm 0.0023$ | $\mathbf{0.9786 \pm 0.0009}$ | $0.8965 \pm 0.0023$ | $\mathbf{0.9795 \pm 0.0009}$ |
| F-MNIST hor. | $\mathbf{0.6751 \pm 0.0042}$ | $\mathbf{0.6728 \pm 0.0038}$ | $0.6274 \pm 0.0041$ | $0.6619 \pm 0.0038$ |
| F-MNIST vert. | $\mathbf{0.8977 \pm 0.0023}$ | $\mathbf{0.8981 \pm 0.0022}$ | $0.7415 \pm 0.0041$ | $0.8921 \pm 0.0024$ |

Table 6: Dissecting out-of-distribution detection with the ELBO. Error estimates obtained through bootstrapping.

## 7  Discussion and conclusion

We have introduced the probabilistic autoencoder, a simple generative model with a lower dimensional latent space that is motivated by probabilistic PCA. Different to variational autoencoders, it builds the probabilistic structure after the first stage of training, but has the advantage of being simple and straightforward to set up and train. Because it is first trained to achieve optimal reconstruction error and then, in a second stage, to produce optimal samples, it performs both tasks reliably and well. We further test its performance in two additional downstream tasks, which we think are particularly relevant for practical applications: anomaly detection and probabilistic image denoising and inputation. We find that the PAE performs all considered tasks at comparable quality as equivalent VAE models, suggesting that ELBO-based variational optimization is not an essential component of this class of models. We find that our proposed OoD metric of NF density in latent space, while a natural byproduct of PAE, can also be used in VAEs if they are complemented with an NF prior.

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

## A PAE on Celeb-A

To demonstrate the feasibility of the PAE approach on higher dimensional, more complex data, we train a PAE on Celeb-A. The celeb-A samples are cropped to the central 128x128 pixels and then downsampled to 64x64 pixels. We used the same preprocessing, architecture and training procedure as for the F-MNIST experiments and a latent space dimensionality of $K = 64$. Samples from this model are shown in figure 5 on the left. On the right we show interpolations between images, produced by projecting two images from the test data set into the latent space of the NF, connecting them in the NF latent space by linear interpolation and sampling along the connecting line in equally spaced intervals. The samples then get forward modeled into data space to produce the images shown. The smooth transitions indicates that the PAE produces a continuous latent space without any holes.

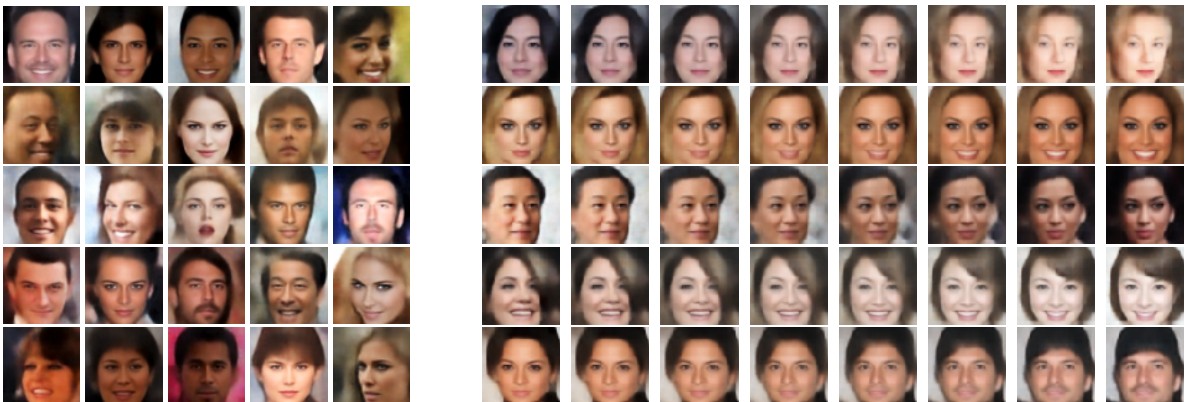

Figure 5: PAE performance on Celeb-A at $K$=64. Samples (left) reach FID=49.2 (reconstructions FID=44.0). Right: Interpolations between samples from the test set.

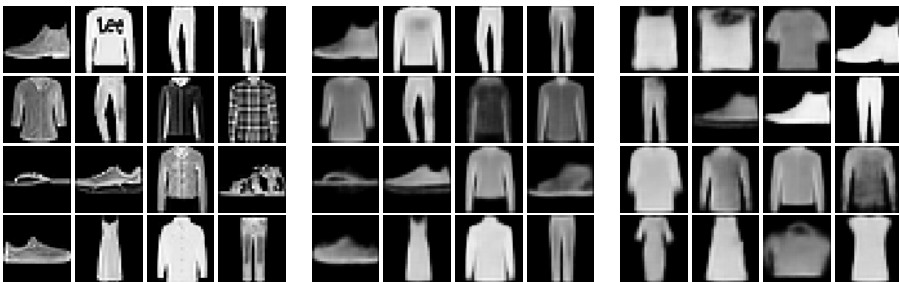

Figure 6: From left to right: Original test images, their reconstructions and samples generated with the vanilla $\beta$-VAE described in Appendix B.

## B  Comparison to vanilla VAE

For completeness, we include here results for a conventional VAE, which shares the same architecture and training parameters as the models used in the other experiments, but uses a standard normal distribution as a prior. We trained the VAE on equation 13 with initial value of $\beta = 100$, which was then linearly annealed during the first 100,000 steps. We show samples and reconstructions of this model in Fig 6 and list reconstruction error and FID scores in Table 7. In all cases the results and visual images are inferior to flow-VAE and PAE described in the main text.

| $\sigma^2_{\text{recon}}$ $[\times10^{-3}]$ ($\downarrow$) | $P_{95\%}(\sigma^2_{\text{recon}})$ $[\times10^{-3}]$ ($\downarrow$) | FID Score ($\downarrow$) |
|---|---|---|
| $12.90 \pm 0.10$ | $68.00 \pm 0.60$ | $32.3 \pm 0.3$ |

Table 7: Reconstruction errors and FID scores of vanilla $\beta$-VAE.

## C  Comparison of anomaly detection with Abati et al.

We compare anomaly detection with the PAE to the results presented in Abati et al. (2019). Their set up is similar, but instead of a two-stage training, they propose a joint training of autoencoder and density estimator. Their loss function is a combination of reconstruction error and estimated density of encoded data. The contribution of the density term to the loss function is controlled by a tunable scalar parameter $\lambda$.

Using the same autoencoder architecture latent space dimension, and batch size[2], we perform anomaly detection experiments as Abati et al. (2019) but with a PAE-style two-stage training: We train models on each class of the CIFAR10 and MNIST datasets separately and then evaluate the outlier detection accuracy when each of these models is applied to the full test dataset containing all classes (considering all other classes outliers). Results of these experiments and the results from Abati et al. (2019) are shown in tables 8 and 9.

| Class | 0 | 1 | 2 | 3 | 4 | 5 | 6 | 7 | 8 | 9 | mean |
|---|---|---|---|---|---|---|---|---|---|---|---|
| Abati et al. | 0.993 | 0.999 | 0.959 | 0.966 | 0.956 | 0.964 | 0.994 | 0.98 | 0.953 | 0.981 | 0.975 |
| PAE | 0.988 | 0.999 | 0.972 | 0.964 | 0.97 | 0.955 | 0.991 | 0.974 | 0.955 | 0.978 | 0.975 |

Table 8: Comparison of MNIST anomaly detection results.

| Class | 0 | 1 | 2 | 3 | 4 | 5 | 6 | 7 | 8 | 9 | mean |
|---|---|---|---|---|---|---|---|---|---|---|---|
| Abati et al. | 0.735 | 0.58 | 0.69 | 0.542 | 0.761 | 0.546 | 0.751 | 0.535 | 0.717 | 0.548 | 0.608 |
| PAE | 0.737 | 0.472 | 0.684 | 0.52 | 0.749 | 0.506 | 0.758 | 0.529 | 0.682 | 0.446 | 0.604 |

Table 9: Comparison of CIFAR anomaly detection results.

## D    Interpolation studies

We expand on the image interpolation results shown in Appendix A by comparing PAE image interpolation with AE image interpolations and adding pixel-level interpolation (linear interpolation between pixel values) as a baseline. Results are shown in figure 7. While both AE and PAE produce fairly well interpolated images of high quality, PAE tends to produce more natural looking interpolations with fewer artifacts.

## E    PAE posterior and application to data inputation

Large-scale data acquisition often results in noisy and incomplete data. In most applications, e.g. when one is interested in finding certain rare features in an image, the aim of data restoration is not only to obtain the most probable uncorrupted image, but also to obtain an estimate of its fidelity. A plethora of generative model based approaches for image reconstruction have been suggested in the literature (Rezende et al., 2014; Mattei & Frellsen, 2018; Dong et al., 2016; Jin et al., 2017; Putzky & Welling, 2017; Ulyanov et al., 2020; Bora et al., 2018; Mattei & Frellsen, 2019), but few of them enable uncertainty quantification (Böhm et al., 2019). With the PAE, we can perform sound posterior-based data restoration. This does not only enable uncertainty quantification, it also provides a framework for consistently including analytical data models, such as physics models.

A latent space posterior for a corrupted data point $\tilde{\boldsymbol{x}} = \boldsymbol{M}\boldsymbol{x}+\boldsymbol{n}$, where $\boldsymbol{M}$ is a pixel-wise mask and $\boldsymbol{n}$ denotes the noise, consists of an implicit likelihood and a prior. The form of the implicit likelihood is determined by the noise properties. For Gaussian noise, $\boldsymbol{n}$, with noise covariance $\boldsymbol{\sigma}_{\text{noise}}$ (typically a diagonal matrix) and a generative model $\boldsymbol{g}_\theta$, trained on uncorrupted data, the implicit likelihood is given by

$$p_\theta(\tilde{\boldsymbol{x}}|\boldsymbol{z}, \boldsymbol{M}, \boldsymbol{\sigma}_{\text{noise}}) = \mathcal{N}\left(\tilde{\boldsymbol{x}}|\boldsymbol{M}\boldsymbol{g}_\theta(\boldsymbol{z}), \boldsymbol{\sigma}_{\text{recon}}^2 + \boldsymbol{\sigma}_{\text{noise}}^2\right). \tag{19}$$

Note that the covariance of this Gaussian likelihood is composed of the generative model's reconstruction error, $\boldsymbol{\sigma}_{\text{recon}}^2$, and the noise level in the corrupted data, $\boldsymbol{\sigma}_{\text{noise}}^2$. For sufficiently high latent space dimensionalities the latter dominates, $\sigma_{\text{recon},i} \ll \sigma_{\text{noise},i}$, ensuring that the likelihood is well approximated by a Gaussian. By replacing $\boldsymbol{x}$ by its generative process, $\boldsymbol{g}_\theta(\boldsymbol{z})$, we bring the inference problem to the low dimensional latent space of the generative model. Posterior analysis of high-dimensional data becomes computationally tractable in this lower dimensional space.

---

[2]obtained from `https://github.com/aimagelab/novelty-detection`

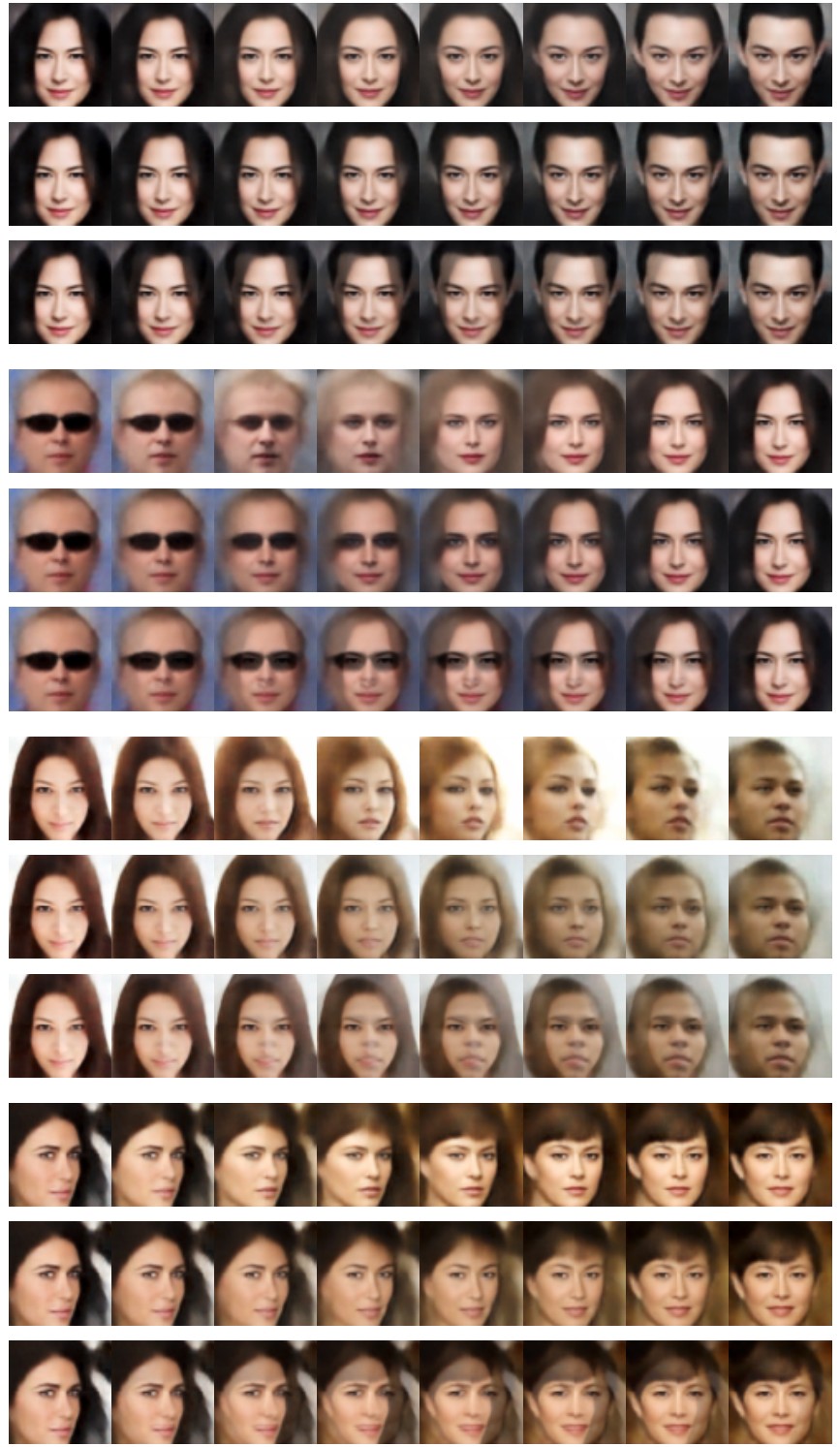

Figure 7: Linear interpolations between images. in each panel we show in the top row the reconstructions from linear interpolations in PAE latent space, in the middle row from AE latent space, and in the bottom row linear interpolations in data space (linear interpolation between pixel values).

The prior is modeled by the generative models' latent space distribution. Combining the two, we obtain the log posterior

$$\ln p_{\theta,\gamma}(\boldsymbol{z}|\tilde{\boldsymbol{x}}, \boldsymbol{M}, \boldsymbol{\sigma}_{\text{noise}}) = \ln p_\theta(\tilde{\boldsymbol{x}}|\boldsymbol{z}, \boldsymbol{M}, \boldsymbol{\sigma}_{\text{noise}}) + \ln p_\gamma(\boldsymbol{z}) - \text{const}. \tag{20}$$

To denoise and inpaint a corrupted image one performs latent space posterior analysis. A point estimate is given by the MAP, $\boldsymbol{z}'$, the maximum of equation 20, which forward modeled into data space, $\boldsymbol{x}_{\text{recon}} = \boldsymbol{g}_\theta(\boldsymbol{z}')$, yields the most likely underlying image. A full posterior analysis can be performed with many techniques, including Laplace approximation, VI or MCMC sampling. Given the multi-modal posterior of some of our examples we fit a full rank Gaussian mixture model to equation 20 following (Seljak & Yu, 2019) in our experiments. We can then sample from this model to obtain other solutions that are compatible with the data.

We examine probabilistic reconstruction with both PAE and flow-VAE on three examples created by corrupting test data with uncorrelated Gaussian noise ($\mu_n$=0, $\sigma_n$=0.1) and masks. The true underlying images are shown in the first column of figure 8 and the corrupted input data to the reconstructions in the second column. The masked areas were chosen to allow for several plausible inpainting solutions. To obtain reconstructions we minimize the negative of the log posterior in equation 20. Since we expect multimodal posteriors we run 20 minimizations starting from random points which we draw from the prior. We keep only those minimization results that are associated with a positive definite Hessian (true minima and no saddle points). In the third and fourth column we show the forward modeled deepest minimum found by this procedure for the PAE (third column) and flow-VAE model (fourth column). The reconstructed examples are well denoised, but flow-VAE and PAE find slightly different inpainting solutions for the masked areas. This is most visible for the first example, where the flow-VAE prefers a clasp on the bag, while the VAE reconstructs a dint at the top. While this suggests that the models have learned slightly different priors and forward models, we find that all reconstructions are very plausible and that the PAE does not seem to perform worse at this task than the flow-VAE. In fact, in the first example, the PAE seems to better reconstruct the area outside of the masked area (e.g. the ribbon on the left), a consequence of the lower reconstruction error of the PAE model.

To obtain uncertainty estimates we fit a Gaussian mixture model using the local minima associated with the largest posterior mass. We use the thus constructed posterior approximation for uncertainty estimation by generating samples from this posterior. The samples are shown in figure 9, with samples from the PAE posterior in the top row and samples from the flow-VAE in the bottom row. We observe some variety in these samples, which we encouraged by posing problems that should have multiple plausible solutions. For example the PAE finds that handbags with different dint depths are compatible with the data. The flow-VAE mostly prefers handbags with clasps, but it also produces two samples without a clasp. It seems that neither of the two models explores the full variety of potential solutions (the flow-VAE missing the dints and the PAE missing the clasps). In the second and third example the PAE model seems to provide some greater variety in the samples. This could be an indicator that the flow-VAE posterior is more complex and wasn't explored properly by our poterior analysis. It is not in scope of this work to explore these possibilities in great detail, instead, we note that flow-VAE and PAE perform comparably well at this task.

## F  Model architectures and training procedures

We use the same encoder and decoder architecture in all experiments. Details are given in table 10 and table 11. The only model parameter that is varied is the dropout rate, which was set to 0.15 in models where it was necessary to avoid overfitting and to zero otherwise. The choice of hyperparameters is described in detail in section 6.3.

The normalizing flow(s) we use are composed of the same building blocks in all experiments but vary in how often the building blocks are repeated. All blocks have the same structure and only differ in the type of transformation that is performed. A block is made up of a transformation of the first half of data variables, a swapping of the first half of data variables with the second half, a transformation on the other half of variables and finally a trainable permutation of the variables. Block 1 applies a neural spline flow transformation, Block 2 a realNVP transformation with shifting and rescaling, Block 3 a realNVP transformation with only

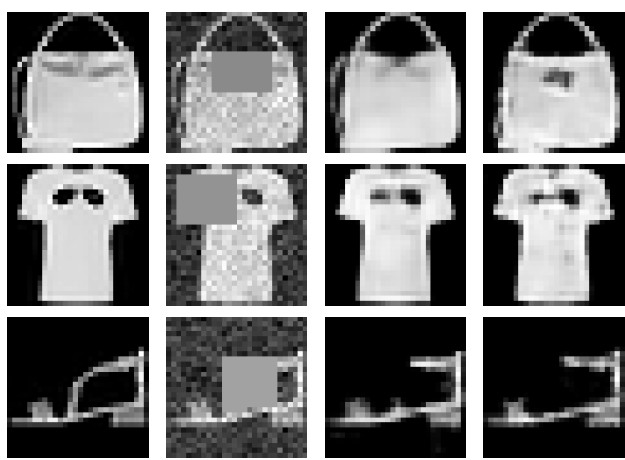

Figure 8: Underlying true data (left column), corrupted data (second column) and most likely reconstructions with the PAE (third column) and flow-VAE (fourth column).

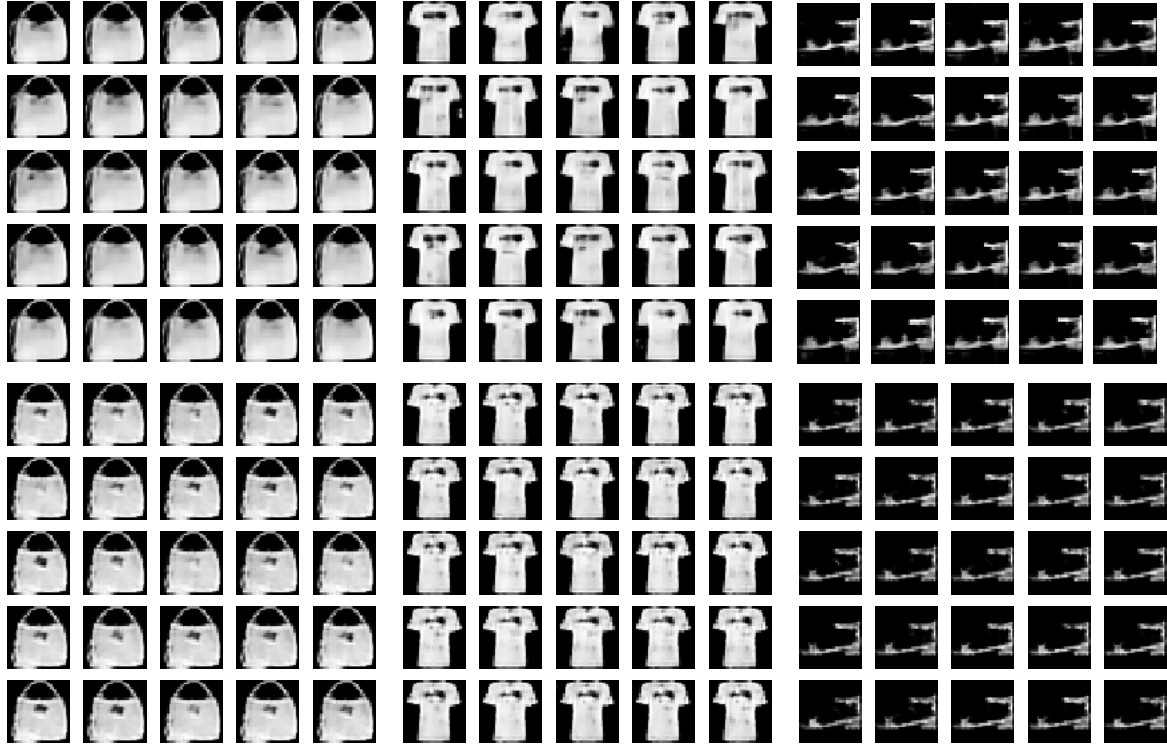

Figure 9: Samples from the reconstruction posteriors (PAE: top row, flow-VAE: bottom row).

**encoder**

| layer | details |
|---|---|
| input | normalized to [-0.5,0.5], de-quantized with uniform noise $\in$ [-1/256, 1/256] |
| convolutional | kernel size=[4,4] , strides=[2,2], filters=64, padding='SAME' |
| leakyReLU | $\alpha$=0.2 |
| convolutional | kernel size=[4,4], strides=[2,2], filters=128, padding='SAME' |
| batch norm | momentum=0.999, $\epsilon$=1e-5 |
| dropout | dropout rate dependent on model, see 2 |
| leakyReLU | $\alpha$=0.2 |
| reshape | flatten |
| linear | output size=1024 |
| batch norm | momentum=0.999, $\epsilon$=1e-5 |
| dropout | dropout rate dependent on model, see 2 |
| leakyReLU | $\alpha$=0.2 |
| linear | 2x latent size |

Table 10: The layout of the encoder network used in all experiments.

**decoder**

| layer | details |
|---|---|
| input | encoded data, latent size=40 |
| linear | output size=1024 |
| batch norm | momentum=0.999, $\epsilon$=1e-5 |
| leakyReLU | $\alpha$= 0.2 |
| linear | output size=128x28/4x28/4 |
| batch norm | momentum=0.999, $\epsilon$=1e-5 |
| leakyReLU | $\alpha$=0.2 |
| dropout | dropout rate dependent on model, see 2 |
| reshape | output size=[28/4,28/4,128] |
| transpose convolution | output shape=[28/2,28/2,64], kernel size =[4,4] , strides=[2,2] |
| batch norm | momentum=0.999, $\epsilon$=1e-5 |
| leakyReLU | $\alpha$=0.2 |
| dropout | dropout rate dependent on model, see 2 |
| transpose convolution | output shape=[28,28,1], kernel size =[4,4] , strides = [2,2] |
| sigmoid | subtract 0.5 to match input data |

Table 11: The layout of the decoder network used in all experiments.

shifting. The architectural details are given in table 12. We use two different layouts in our experiments, a deeper flow with [2x block 1, 4x block 2 and 4x block 3] and a simpler flow with [1x block 1, 2x block 2 and 1x block 3]. For our Celeb-A experiments, we use a normalizing flow consisting of [1x block 1, 2x block 2 and 2x block 3]. Every flow also features a re-scale operation that ensures that the encoded training samples lie within the range $z_i \in [-1, 1]$.

The neural spline flow uses two networks to determine the bin widths and slopes of the rational quadratic splines. The layouts of these networks are listed in table 13.

### normalizing flow

| blocks | layer | details |
|---|---|---|
| | rescale | scale dependent on model, see 14 |
| 1 | neural spline transform | bins=36, bins: bin network, slopes: slope network |
| | swap permutation | swap first half of dimensions with second half |
| | neural spline transform | bins=36, bins: bin network, slopes: slope network |
| | trainable permutation | GLOW-style LU decomposition of orthogonal matrix |
| 2 | real NVP transform | trainable shift and rescale |
| | swap permutation | swap first half of dimensions with second half |
| | real NVP transform | trainable shift and rescale |
| | trainable permutation | GLOW-style LU decomposition of orthogonal matrix |
| 3 | real NVP transform | trainable shift |
| | swap permutation | swap first half of dimensions with second half |
| | real NVP transform | trainable shift |
| | trainable permutation | GLOW-style LU decomposition of orthogonal matrix |
| | rescale | 1/scale dependent on model, see table 14 |

Table 12: Building blocks of the normalizing flows used in all experiments.

### slope network

| layers | details |
|---|---|
| dense | output size=latent size//2 |
| leakyReLU | $\alpha$=0.2 |
| dense | output size=bins-1 |
| reshape | [latent size/2, bins-1] |
| softplus | softplus(x)+1e-2 |

### bin network

| layers | details |
|---|---|
| dense | output size=latent size/2 |
| leakyReLU | $\alpha$=0.2 |
| dense | output size=latent size/2 |
| leakyReLU | $\alpha$=0.2 |
| dense | output size=bins |
| reshape | [latent size/2, bins] |
| softmax | softmax(x)(2-1e-2 bins)+1e-2 |

Table 13: Networks used to determine bin widths and slopes in the rational quadratic spline of the neural spline flow.

## G  Flow-VAE ablation studies

We ran several flow-VAE models with different values of $\sigma$, with $\sigma$ being the scale parameter in the implicit ELBO likelihood, a multivariate Gaussian with $\boldsymbol{\mu}=\boldsymbol{f}_\theta(\boldsymbol{z})$ and $\boldsymbol{\Sigma}=\sigma\mathbb{1}$,

$$p_\theta(\boldsymbol{x}|\boldsymbol{z}) = \mathcal{G}(\boldsymbol{x}|f_\theta(\boldsymbol{z}),\sigma\mathbb{1}). \tag{21}$$

The objective of this ablation study is to test whether the flow-VAE performance depends on this parameter. Similar to the $\beta$-parameter, $\sigma$ could have an influence on the relative contribution of distortion and rate term to satisfying the ELBO objective during training. The results of this ablation study (in terms of reconstruction error and sample quality) are listed in table 15. We do not find a strong dependence of neither reconstruction error nor sample quality on $\sigma$. Somewhat counter-intuitively we get slightly higher reconstruction errors for lower values of $\sigma$, an indication that the model is starting to overfit. A test run with $\sigma = 0.05$ (not listed in the table) resulted in catastrophic overfitting. A slightly higher value, $\sigma = 0.1$, seems to be a sweet spot, with the lowest reconstruction error, no signs of overfitting and good FID score. This is the model we use in the result section. We note that the other values of $\sigma$ we tried would not change our results in a way that invalidates our comparison or conclusions. We also note that such an ablation study/ parameter optimization is not needed for the PAE training, which highlights an advantage of the PAE approach.

**normalizing flow layouts for each model**

| model name | $\beta_0$-VAE | PAE & flow-VAE | flow-VAE(s) |
|---|---|---|---|
| repeats flow block 1 | 2 | 2 | 1 |
| repeats flow block 2 | 4 | 4 | 2 |
| repeats flow block 3 | 4 | 4 | 1 |
| flow rescale scale | 344 | 3.44 | 1 |

Table 14: Normalizing flow layouts of different models. Blocks are described in table 12.

**$\sigma$-ablation study**

| $\sigma$ | 0.08 | 0.1 | 0.12 |
|---|---|---|---|
| $\overline{\sigma^2_{\text{recon}}}$ $[\times 10^{-3}]$ ($\downarrow$) | $7.85 \pm 0.09$ | $\mathbf{6.22 \pm 0.07}$ | $8.35 \pm 0.09$ |
| $P_{95\%}(\sigma^2_{\text{recon}})$ $[\times 10^{-3}]$ ($\downarrow$) | $36.5 \pm 0.3$ | $\mathbf{35.7 \pm 0.4}$ | $39.9 \pm 0.5$ |
| sample FID score ($\downarrow$) | $\mathbf{22.5 \pm 0.2}$ | $22.8 \pm 0.3$ | $\mathbf{22.8 \pm 0.3}$ |

Table 15: Flow-VAE ablation study: reconstruction error and sample quality as a function of $\sigma$, the width of the Gaussian likelihood entering the distortion term in the ELBO.

