# OpenReview forum: "Probabilistic Autoencoder"
_TMLR — Accepted by TMLR_

### Review · Reviewer_kA9U · 2022-06-27

**Summary Of Contributions:**

This work proposes probabilistic Autoencoder (PAE), a simple procedure motivated from probabilistic PCA. It is comprised from two stages; on the first stage a standard deterministic Autoencoder is trained on a data set and on the second stage a normalising flow prior is trained to obtain a density over the latent space of the encoder. Armed with this normalising flow prior, the authors propose to use PAE for several tasks associated usually with generative models. The authors also propose a new metric for out-of-distribution detection, namely the log probability of the latent code under the (learned) prior. There are relatively extensive ablation studies and experimental evaluation. Therefore, the contributions of this work are:

1. The PAE 2-stage framework, which is simple and seems to work reasonably well in practice
2. The new out-of-distribution metric which relies on density estimation in the latent space
3. The several experiments done to compare PAE to VAE in a variety of tasks

**Broader Impact Concerns:**

No concerns.

**Requested Changes:**

1. The authors argue about the compression properties of the PAE model, but no experiments are done to show its performance. Given that they train the normalizing flow prior, I would have expected at least an estimate of the log-likelihood of the test data. The latter would serve as a reasonable proxy for the compression capabilities of PAE versus, e.g., the VAE baseline. This is a critical experiment for the data compression claim (i.e., the one at the conclusion).
2. Given the similarity with RAEs (Ghosh et al., 2020), I would have expected that the authors did a comparison against that method as well. This would highlight whether a more flexible latent density is needed over the simpler mixture of Gaussians baseline. I would say that this is a medium importance point, given that there is also some novelty on the latent space OOD metric of this work.
3. The data modelled in this work are discrete / have bounded support, however a Gaussian likelihood is assumed for both the VAE and PAE. What happens when a more appropriate likelihood is used? For example, on MNIST / FMNIST, you could try binarising the data and using a Bernoulli likelihood, or you could use, e.g., a discretised logistic [1] on the pixel values directly as they are 8-bit grayscale values [2]. This is not a critical experiment, but it would strengthen this work.
4. I would appreciate if the authors discuss how is the interpolation behaviour of the standard AE that the PAE is based upon. This would highlight whether the normalising flow “fixes” potential holes in the latent space of the AE, or whether these holes are not there to begin with (and thus the normalising flow does not bring much extra compared to the standard AE). This is not a critical experiment, but it would strengthen this work.

Minor points
- In eq. 9, I take that the $p(z)$ refers to the empirical distribution of the latent codes. I would advise that the authors change this to, e.g., $\tilde{p}(z)$ in order to distinguish it from the latent space prior (which is also denoted as $p(z)$).
- It would be good if the authors mention how the VAE is evaluated for all the metrics. Are they averaged metrics over multiple samples from $q(z|x)$ or is only the mean of the variational posterior propagated (i.e., which is closer to what a standard AE does)?
- At page 10, in the first sentence after “Outlier detection accuracy”, the authors mention the “outlier detection metrics listed in 1”; what is “1”?

[1] PixelCNN++: Improving the PixelCNN with Discretized Logistic Mixture Likelihood and Other Modifications, Salimans et al., 2017

[2] Fashion-MNIST: a Novel Image Dataset for Benchmarking Machine Learning Algorithms, Xiao et al., 2017


**Strengths And Weaknesses:**

Strengths:
- Simple approach that works well in practice
- Well written manuscript that is easy to follow
- Good experimental evaluation
- Thorough explanation of the motivation and extensive ablation studies
- New metric for OOD works quite well and can be applicable to VAEs as well

Weaknesses:
- Incremental given other works in the field; this two step procedure has been explored before, e.g., in VQ-VAEs (Razavi et al., 2019) and RAEs (Ghosh et al., 2020). The main difference with, e.g., RAEs, seems to be that the authors consider a more flexible latent density model (i.e., a normalizing flow) instead of a mixture of Gaussians.
- Some claims are not backed up by experimental evaluation (e.g., compression capabilities)

---

### Review · Reviewer_RVev · 2022-07-06

**Summary Of Contributions:**

The paper proposes an approach to generative modeling, which consists of two stages:
1. Train a deterministic autoencoder model, minimizing the reconstruction error
2. Train a normalizing flow model to estimate a density in the autoencoder embedding space.

The authors show that this approach is competitive with VAEs of the same architecture, where the flow serves as a prior in the latent space.

**Broader Impact Concerns:**

No concerns.

**Requested Changes:**

**Important**:
- Could you add a baseline using a VAE with a regular Gaussian likelihood and the same encoder-decoder architecture to the experiments?

Generally, I would feel much more comfortable recommending acceptance if the authors could demonstrate that PAEs are useful relative to other generative models, and not just comparable to specifically VAEs with NF priors. Is there an application where you would expect PAEs to be particularly useful?

**Typos and minor suggestions**:
- In Eq. (1) you say that $O \in R^{N \times K}$, $K < N$, and $O O^T = 1_N$. Wouldn't $O O^T$ be low-rank, as $K < N$?
- "this is assumption is not always satisfied" (page 5)
- "lowest training loss on validation data" (page 9)

**Strengths And Weaknesses:**

**Strength**: The paper proposes a fairly simple approach and shows that it is competitive with VAEs. In particular, the proposed approach does not use any kind of variational inference.

**Weakness?** The main advantage of the proposed approach appears to be simplicity. However, simplicity is debatable. In particular, unlike VAEs the proposed method requires a two-stage approach, where the components of the model are trained sequentially. The VAE model is trained jointly with a single optimization process. Moreover, standard VAEs with a Gaussian latent prior (not using NFs) are arguably simpler than the proposed model.

**Strength:** The experiments are done carefully. In particular, the authors present a very detailed description of the hyper-parameters and the tuning procedure. The authors developed a non-trivial $\beta_0$-VAE baseline to ensure a fair comparison.

**Weakness:** Generally, the experiments are not particularly conclusive. On the reconstruction error, PAE achieves the best results (which is expected given that unlike the baselines PAE does not include regularization, e.g. KL or noise in the encoder). However, flow-VAE achieves better FID scores.

**Strength:** The proposed method achieves good out-of-distribution detection performance, improving upon VAE baselines.

**Weakness:** The OOD detection experiments are somewhat limited, as the authors only consider the FashionMNIST in-distribution dataset. Moreover, multiple prior works also indicated that normalizing flows applied to embeddings can achieve strong OOD detection results [e.g. 1-3]. Moreover, it is not clear how PAE compares to specialized OOD detection methods.

Generally, the main takeaway from the paper is that PAE performs comparably to VAEs with NF latent space prior, while arguably being simpler. While this is an interesting result, it is not clear whether PAEs will be useful in practice, compared to other generative models and even VAEs with regular Gaussian priors.


[1] *Deep Residual Flow for Out of Distribution Detection*; Ev Zisselman, Aviv Tamar

[2] *Why Normalizing Flows Fail to Detect Out-of-Distribution Data*; Polina Kirichenko, Pavel Izmailov, Andrew Gordon Wilson

[3] *Hybrid Models for Open Set Recognition*; Hongjie Zhang, Ang Li, Jie Guo, Yanwen Guo

---

### Review · Reviewer_3jv7 · 2022-07-09

**Summary Of Contributions:**

This submission presents Probabilistic Autoencoder (PAE), a latent-variable generative model trained in a two-stage process. In the first stage an autoencoder is trained with a simple reconstruction objective. In the second stage a flow-based model is used to estimate the density of input data that has been encoded in latent space. The model is compared to a variety of VAE-based baselines in reconstruction, generative modelling, and anomaly detection tasks on image datasets.

The notion of training a prior over latent variables after first training an auto-encoder has been introduced previously (e.g. in VQ-VAE [1]), but this paper provides a detailed explanation of the process, as well as a range of experiments that reveal the experimental benefits of the approach compared to more typical VAE-style baselines.

Using probabilistic PCA as a motivating example, the paper explores the impact of latent dimensionality on anomaly detection, both using data and latent space density estimation-based detectors.  Again, this is not the first example of latent-space based anomaly detection (e.g. [2]), but the connections to PPCA, and the empirical exploration in this context are useful contributions.

[1] Van Den Oord, Aaron, and Oriol Vinyals. "Neural discrete representation learning." Advances in neural information processing systems 30 (2017).
[2] Abati, Davide, et al. "Latent space autoregression for novelty detection." Proceedings of the IEEE/CVF conference on computer vision and pattern recognition. 2019.

**Broader Impact Concerns:**

I think the ethical implications of the work are quite limited and don't warrant a broader impact statement. The work presents a generative image model and it inherits the ethical concerns for this application such as privacy (deep-fakes) and emebdded bias. However it is far from a state of the art model, and is unlikely to be applied in its proposed form for any harmful purpose.

**Requested Changes:**

* [strengthen] An expansion of the discussion in the PPCA-based anomaly detection section to clarify statements like "this noise component is uninformative for the outlier detection task and might obscure the results".
* [strengthen] An more thorough explanation of how the PPCA anomaly detection results motivate the use of latent spaces for anomaly detection. (See the point in the Weaknesses for more info)
* [critical] A comparison to existing methods in the literature for anomaly detection. In particular to "Latent space autoregression for novelty detection." At the very least this method should be discussed, and ideally the authors would report AUROC scores for their methods on the MNIST and CIFAR eval regimes described in that paper. Otherwise we simply don't know how useful the proposed method is in relation to other existing methods.



**Strengths And Weaknesses:**

Strengths:

* The paper explores the impact of using a two-stage training process for VAE-style models . While other works have proposed similar training approaches, it is useful to directly compare the two-stage process to the more standard joint training with the ELBO objective in a controlled setting.
*  The paper also shows how such a model can be used for anomaly detection, by doing density estimation either in latent space, or in data space. The experiment and analysis with PPCA motivates the use of latent space based anomaly detection. In particular the line "The very small eigenvalues are likely to be mis-estimated and not informative for the outlier detection task" seems potentially quite insightful (but should be expanded for clarity).

Weaknesses:

* "this noise component is uninformative for the outlier detection task and might obscure the results" - This is a strong statement that makes some underlying assumptions about the nature of outliers. Maybe it is a good assumption, but that should be made explicit.
* "In our experiments, we find that the PAE latent space density is a superior anomaly detector than the ELBO" - because of the max value of the AUROC curves? The difference between AUROC = 0.980 and AUROC = 0.974 is pretty tiny. It does seem like the p(z) anomaly detector is more robust to the number of PCA components, but this is not discussed in the text.
* Anomaly detection: while the experiments are interesting, I wanted to see more discussion of other anomaly detection methods. The paper answers the question of: Which AE-style model (and which terms in the ELBO) is more effective as an anomaly detector, but it doesn't say in how useful the best presented anomaly detector is compared to other methods in the literature. For those who want to build the most effective anomaly detectors this is an important question to answer.
* "test data set into the latent space of the NF, connecting them in the NF latent space by a straight line and
sampling along this line in equally spaced intervals" - I'm unsure whether this refers to a simple linear interpolation in the latent space, or in a way that somehow reflects the probability density associated with the NF (e.g. projected onto a manifold of high probability latents). I think this isn't quite clear enough.
* "Comparing the PAE to the β0-VAE tests whether training on the reconstruction error instead of the distortion term
offers any advantage" - there is an additional confounding factor which is that the beta0-VAE samples from the inference distribution q(z ; x) with a fixed variance, wheras PAE deterministically regresses a point z given x. This extra noise will typically incentivize the model to "spread out" z-space more, and may impact flow-based density estimation in the second stage of training.
* If the intent with B0-VAE is to "tests whether training on the reconstruction error instead of the distortion term offers any advantage", then why not just replace the reconstruction term in PAE? Although it shouldn't make any difference as the Gaussian reconstruction log-likelihood (with fixed variance) is equal up to a constant to the MSE used to train the PAE.
* The lack of discussion or comparisons to "Latent space autoregression for novelty detection." [1] is a bit of a problem, as it is a well cited paper that proposes a similar strategy for novelty detection.

[1] Abati, Davide, et al. "Latent space autoregression for novelty detection." Proceedings of the IEEE/CVF conference on computer vision and pattern recognition. 2019.

---

### Decision · Action_Editors · 2022-09-13

**Recommendation:** Accept with minor revision

**Comment:**

The paper explores a two-step approach to generative modelling: first a non-linear autoencoder is trained to reconstruct the data, then a density model (here a normalizing flow) is trained to model the latent-space distribution. This constitutes an alternative to variational autoencoders (VAEs), and the paper conducts a thorough comparison between the two.

Although the two-step approach is not new (for example VQ-VAE [1] and RAE [2] follow it too), the paper provides a number of useful insights and a thought-provoking and careful comparison to VAEs. After some requested additional experiments that the authors supplied during the discussion period, the reviewers are satisfied that the paper's claims are adequately supported by empirical results. Therefore, the paper meets the criteria for acceptance to TMLR.

[1] van den Oord et al., 2017; Razavi et al., 2019

[2] Ghosh et al., 2020

Nonetheless, the paper requires a minor revision to rephrase the main claims and bring them in line with what the experiments have demonstrated. In particular, the experiments substantiate the claims that the proposed approach has advantages compared to equivalent VAE models, but they don't support claims that the proposed model achieves state-of-the-art sample quality or outlier-detection performance. Therefore, the following statements in the abstract must be rephrased:

"the PAE [..] produces state-of-the-art sample quality"

"[...] achieves state-of-the-art results and outperforms other likelihood-based estimators"

Additional minor points to consider revising:
- In Appendix B, "batch sizefootnote" looks like a typo.
- Appendix B contains important results that would be worth highlighting in the main body (either by moving them to the main body or linking to them from the main body).